# From imaging to computational domains for physics-driven molecular biology simulations: Hindered diffusion in platelet masses

Catherine House[1☯], Ziyi Huang[1☯], Kaushik N. Shankar[1], Sandra J. Young[2], Meghan E. Roberts[3], Scott L. Diamond[1], Maurizio Tomaiuolo[4], Timothy J. Stalker[3], Lu Lu[5], Talid Sinno[1]*

**1** Department of Chemical and Biomolecular Engineering, University of Pennsylvania, Philadelphia, Pennsylvania, United States of America, **2** Independent Researcher, Philadelphia, Pennsylvania, United States of America, **3** Cardeza Foundation for Hematological Research, Department of Medicine, Thomas Jefferson University, Philadelphia, Pennsylvania, United States of America, **4** Department of Pediatrics, Children's Hospital of Philadelphia, Philadelphia, Pennsylvania, United States of America, **5** Department of Statistics and Data Science, Yale University, New Haven, Connecticut, United States of America

☯ These authors contributed equally to this work.
* talid@seas.upenn.edu

## Abstract

When formed in vivo, murine hemostatic thrombi exhibit a heterogeneous architecture comprised of distinct regions of densely and sparsely packed platelets. In this study, we utilize high-resolution electron microscopy alongside machine learning and physics-based simulations to investigate how such clot microstructure impacts molecular diffusivity. We used Serial Block Face – Scanning Electron Microscopy (SBF-SEM) to image select volumes of hemostatic masses formed in a mouse jugular vein, producing high-resolution 2D images. Images were segmented using machine learning software (Cellpose), whose training was augmented by manually segmented images. The segmented images were then utilized as 2D computational domains for Lattice Kinetic Monte-Carlo (LKMC) simulations. This process constitutes a computational pipeline that combines purely data-derived biological domains with physics-driven simulations to estimate how molecular movement is hindered in a hemostatic platelet mass. Using our pipeline, we estimated that the 2D hindered diffusion rates of a globular protein range from 2% to 40% of the unhindered rate, with denser packing regions lending to lower molecular diffusivity. These data suggest that coagulation reactions rates, thrombin generation and activity, as well as platelet releasate activity may be drastically impacted by the internal geometry of a hemostatic thrombus.

## Author summary

Hemostasis and coagulation are two exquisitely complex, intertwined, and tightly regulated biological processes. Dysregulation of either process may lead to

**Data availability statement:** Raw images, manually segmented images, and LKMC code for diffusion calculations are available at https://github.com/HTM-Sinno-Stalker/Hindered-Diffusion-in-Platelet-Masses

**Funding:** CH was supported by the Hematology NHLBI T32 training grant T32-HL007439, National Institutes of Health (https://www.nih.gov/). TS and MT were supported by grant R21-HL153946-02 from the NHLBI, National Institutes of Health (https://www.nih.gov/). TJS was supported by grants P01-HL146373 and R35-HL150818 from the NHLBI, National Institutes of Health (https://www.nih.gov/). LL was supported by Grant No. DMS-2347833, National Science Foundation (https://www.nsf.gov/). The funders had no role in study design, data collection and analysis, decision to publish, or preparation of the manuscript.

**Competing interests:** The authors have declared that no competing interests exist.

severe health consequences or death. Coagulation reactions have been extensively studied under static laboratory conditions, which are different from *in vivo* conditions. It is therefore imperative to understand if and how the chemical reactions underlying coagulation are regulated by the environment where they occur. *In vivo* experimentation enables us to image hemostasis, but not chemical reactions. Physics-driven molecular simulations of chemical reactions can bridge the gap, provided the physical environment is correctly represented computationally. The present work serves as a much-needed foundation for image-to-computation for physics based molecular simulations in biological domains.

## Introduction

Previous studies [1–7] have suggested that the tortuous and platelet-rich internal structure of a hemostatic thrombus may be an important factor contributing to the regulation of essential chemical reactions in both hemostatic and thrombotic responses. Specifically, regions of densely packed platelets are characterized by pore spaces estimated to be as small as 20–30 nm in diameter or less [8], which is on the same scale of some plasma proteins (~10 nm in diameter). Within these narrow pore spaces, it is hypothesized that molecular diffusion and reactions may be slowed due to the physical hindrance posed by platelets, fibrin, and other cells. Consequently, these regions may represent islands of chemical stagnation and bottlenecks for chemical signal propagation during hemostasis and/or thrombosis.

Recent efforts led by different groups have provided glimpses into the internal microstructure of thrombi formed in multiple physiologic contexts, albeit with a focus on overall biological characteristics such as cellular content, distribution, and spatial arrangement [9–12]. While these studies have provided valuable insights into the structural and biochemical heterogeneities of a thrombus and how they relate to spatio-temporal evolution, they are generally not sufficiently spatially resolved to address the coupling between structure and molecular diffusion. Some prior computational studies have specifically attempted to describe intrathrombus molecular transport using various approaches. For example, [4,5] used assemblies of ellipsoidal particles in 2D and 3D, respectively, to generate idealizations of a thrombus microarchitecture with different gap size distributions. Diffusive and convective transport was then assessed as a function of gap size distributions. A different approach was taken by Voronov et al. [1] whereby intravital confocal microscopy was used to image the intrathrombus microenvironment and the data used to construct a 3D computational domain for performing fluid flow simulations. Notably, these studies have not yet included a fully resolved description of the pore network microarchitecture, which is needed to achieve a quantitative understanding of whether and how molecular diffusion and reaction rates are impacted by structure in a biological domain. To accomplish this objective, we require: 1) volume imaging at a resolution that is appropriate for investigating protein-protein and cell-protein interactions (i.e., approaching a few nanometers), 2) the ability to transform the imaging, with minimal bias, into a computational domain, and 3) particle-resolved simulations of molecular diffusion and reaction.

PLOS Computational Biology

Here, we present a novel pipeline combining 1) electron microscopy to generate high-resolution images of intrathrombus environments, 2) artificial intelligence techniques for image segmentation into computational domains, and 3) multiscale particle simulations based on the lattice kinetic Monte Carlo technique to assess molecular diffusion as a function of thrombus microstructure; see Fig 1. Imaging was based on Serial Block Face Scanning Electron Microscopy (SBF-SEM), which generates high-resolution, sequential co-linear stacks of 2D images from selected regions of hemostatic masses formed *in vivo* in a mouse jugular vein. While these image stacks may, in principle, be used to create 3D domains, here we restrict our analysis to 2D by considering a single image at a time. A manually labelled portion of these images was used to train Cellpose, a freely available deep-learning software for object detection and image segmentation of cellular configurations [13,14]. Segmentation of SBF-SEM images poses specific challenges, primarily due to the intricate and overlapping structural characteristics of biological tissues. These factors make it difficult for traditional image processing algorithms to achieve accurate segmentation. Deep learning models, especially those utilizing U-Net architectures like Cellpose, have shown great promise in this regard [13]. Prior studies have demonstrated the effectiveness of U-Net in segmenting a variety of biomedical images, including electron microscopy data, validating its application in complex segmentation tasks of noisy or variable-quality images [15, 16].

The success of the segmentation procedure was analyzed by comparison to additional manually labelled images not used for training. The segmented images were then binarized into domains suitable for performing particle-based lattice kinetic Monte Carlo (LKMC) simulations of intrathrombus diffusion. Importantly, we show that the binarization threshold

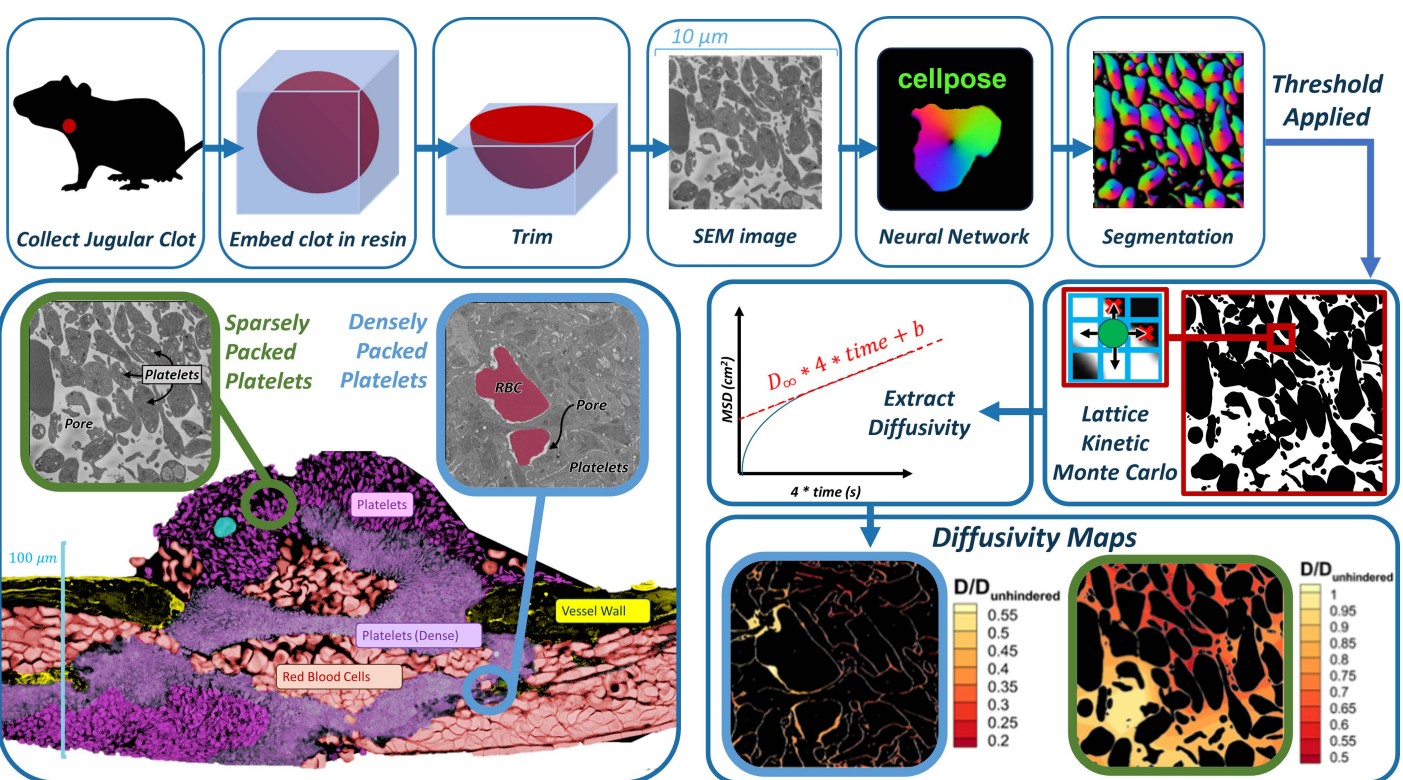

**Fig 1. Overall experimental-computational workflow for image-driven simulation of molecular diffusion in the intrathrombus environment.** Top row, left to right: Hemostatic clots are generated in mouse models, extracted, processed, imaged at the single-platelet resolution with electron microscopy, followed by image segmentation with a trained convolutional neural network model. Middle row, left to right: images are binarized and used to construct simulation domains for lattice kinetic Monte Carlo simulations of hindered molecular diffusion. Bottom row: detailed spatially resolved maps of diffusivity are constructed. Bottom left image shows highly heterogenous intrathrombus environment.

serves as an adjustable parameter that strongly impacts the results of the LKMC simulations. We introduce an additional 'functional validation' strategy by comparing diffusivities predicted in machine learning (ML)-segmented domains to those in manually labelled domains and show that a single robust binarization strategy can be defined.

Using our validated workflow, we find that densely packed regions within a thrombus can reduce diffusivity up to two orders-of-magnitude for a spherical molecular entity of size ~200–400 kDa (~ 5–8 nm radius) as compared to diffusion in an unhindered environment. Moreover, we demonstrate that the present approach enables a functional comparison of diffusion across different thrombus environments that appear similar. That is, two segmented images or domains may 'look' similar but may exhibit different diffusivities due to subtle variations in physical characteristics of the pore network. Although we do not explicitly consider the coagulation cascade in the present work, the present approach is readily augmented to include reaction networks within the LKMC framework to provide a more explicit picture of how knowledge of the physical environment may enable a better understanding of biological responses in tissues.

## Materials and methods

### Ethics statement

The authors have declared that no competing interests exist. All animal studies were approved by the Institutional Animal Care and Use Committee of Thomas Jefferson University.

### Mice and reagents

Male C57Bl/6 mice (Jackson Laboratories, Bar Harbor, ME), 8–12 weeks old, were used for all studies. Paraformaldehyde, glutaraldehyde, and sodium cacodylate were purchased from Electron Microscopy Sciences.

### Mouse large vessel puncture injury and serial block face-scanning electron microscopy

The right external jugular vein of anesthetized mice was isolated and gently cleaned of connective tissue. A puncture injury was created in the vessel using a sterile 160 μm diameter needle. In all cases the puncture injury resulted in bleeding. Extravasated blood was continually rinsed away from the injury site by slow superfusion of normal saline delivered via a syringe pump. Hemostasis was achieved in approximately 1 minute. The experiments were stopped 5 minutes post-injury via transcardiac perfusion of sodium cacodylate buffer (0.2 M sodium cacodylate, 0.15 M sodium chloride, pH 7.4), followed by perfusion of 2% paraformaldehyde/2% glutaraldehyde to fix tissues. After perfusion, the punctured blood vessel was excised, placed in a 35 mm dish coated with silicone and submerged in 2% paraformaldehyde/2% glutaraldehyde. The vessel was then carefully cleaned of any remaining connective tissue, cut along its length, opened, and pinned to the silicone pad with the intraluminal portion of the vessel face up. The vessel was shipped in fixative on ice to the Mayo Clinic Microscopy and Cell Analysis Core for SBF-SEM imaging. Samples were prepared for SBF-SEM imaging as described in [11]. Imaging was performed using a VolumeScope SBF-SEM (ThermoFisher, Waltham, MA). High resolution serial images were captured at 8 nm *x/y* pixel resolution with 50–100 nm *z*-plane spacing.

### Image segmentation

Segmentation of complex images with variable quality and resolution, such as those obtained by SBF-SEM imaging, is an inherently challenging task. But deep learning models utilizing convolutional neural networks have been shown to be effective in segmenting a variety of biomedical images, including electron microscopy data, as well as in image-based disease identification [16–19]. Moreover, the adaptability of deep learning to various imaging modalities and its capacity for feature extraction even in noisy or variable-quality images [15, 16] position it as an ideal tool for SBF-SEM image segmentation. Despite these successes, the acquisition of large, *annotated* datasets for training deep learning models is often costly, time-consuming and represents a significant bottleneck [20]. Transfer learning, which refers to transferring

knowledge gained from one task (often with abundant data) to another related but distinct task (where data is limited) [21,22], offers a powerful solution. In transfer learning, deep learning models are first trained on large and diverse data-sets, to produce pre-trained models. The models are then fine-tuned with additional training to a more limited, but specifically relevant data set. Transfer learning has recently been employed in various medical research fields such as medical image segmentation [23], cell group segmentation in cancer or disease areas [24,25], cell nucleoplasm segmentation [26], and glucose levels prediction in diabetes patients [27]. Transfer learning not only accelerates the training process, but also improves the generalization capability of the model, including scenarios with significant variations in image quality and structure. This adaptability is particularly beneficial in analyzing the intricate structures of hemostatic thrombi in SBF-SEM images, where conventional training methods might falter due to data scarcity.

In the present study, segmentation of SBF-SEM images was carried out using Cellpose software [13,14], a deep learning-based generalist model for cellular segmentation that takes advantage of the transfer learning concept; see Fig 2. Cellpose is based on a modified U-Net network pre-trained with a broad range of cell types and staining methods whose diversity collectively provides a robust foundation for image segmentation of cellular images. These include 100 fluorescent images of cultured neurons with cytoplasmic and nuclear stains, 216 images with fluorescent cytoplasmic markers, and a collection of images from various microscopy methods and public sources. Cellpose also allows for additional training on user-specific datasets, which in the present work are manually annotated SBF-SEM images of thrombi.

Once trained, the Cellpose network predicts two output fields for a given image: a likelihood field of a pixel being part of a cell ('cell probability') and a vector gradient field ('gradient flow'), which together yield the final segmented image, as shown in Fig 2. The cell probability output is computed through a sigmoid function, producing values that range from 0 to 1, allowing for a probabilistic interpretation of cellular presence within the image. The gradient flow is used to enhance cellular segmentation and includes two vector fields that guide pixels towards the center of each cell, not necessarily directly,

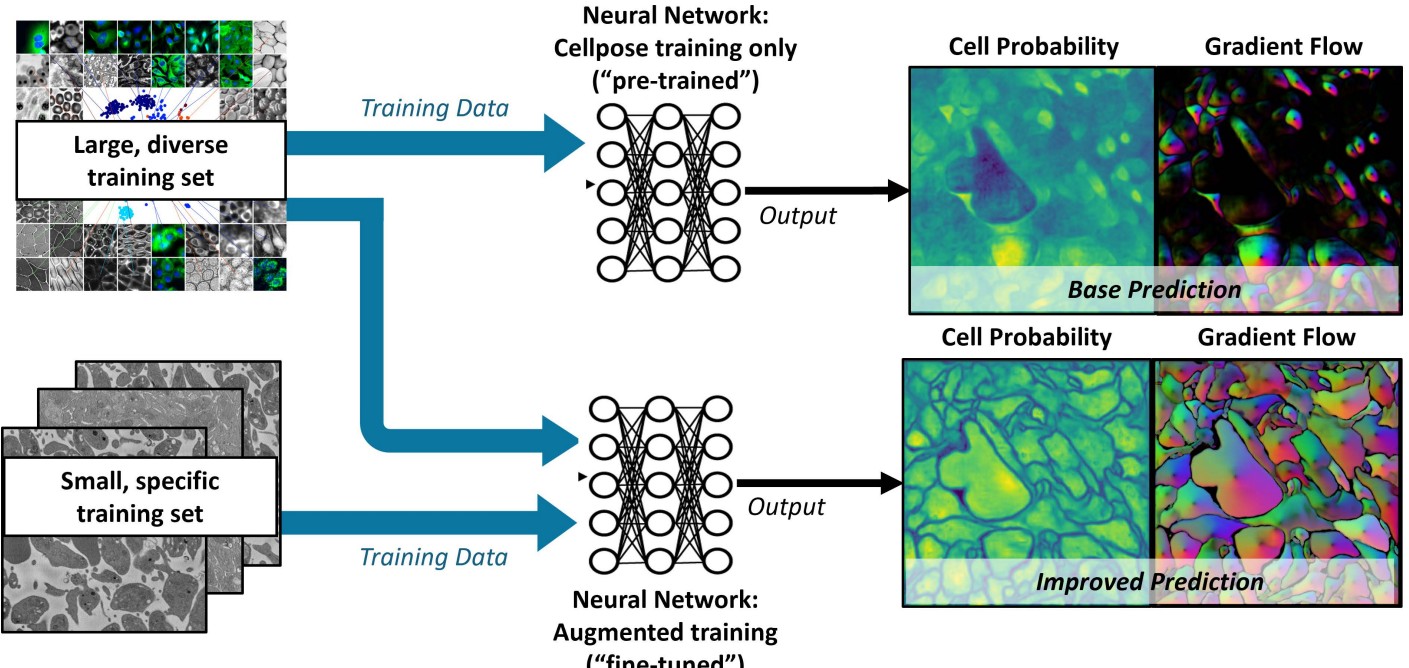

**Fig 2. Cellpose model architecture.** A convolutional neural network (CNN) is trained to predict a map of cell probability and a map of gradient flow [13,14]. Top row: a pretrained model is based on a diverse body of images but not specifically trained on thrombus images. Bottom row: fine-tuning of the model retains the pretraining and only requires few additional thrombus-specific images to greatly improve model performance.

but through iterative translations that eventually converge. This method, akin to heat diffusion simulation, assigns a heat source at the cell's calculated center-of-mass and iteratively distributes this value to the surrounding pixels, refining the segmentation map. If needed, Cellpose also combines the cell probability and gradient flow maps into a final segmented image, or 'mask'; however, this final step requires the specification of additional threshold parameters. In the remainder of the paper, masks are used only for visualization of segmentation results; diffusion analysis is performed directly on either the cell probability or the gradient flow outputs.

## Lattice kinetic monte carlo simulations

The kinetic Monte Carlo method refers to a class of simulation techniques for evolving the configuration of a system governed by stochastic dynamics. It may be considered as a spatially resolved variation of the Gillespie algorithm [28,29] that is commonly employed for evolving reaction networks. The lattice variant of KMC, or LKMC, restricts the spatial coordinates to a predefined lattice or grid and is most commonly applied to diffusion-driven phenomena but can also be readily extended to accommodate advection and interparticle interactions [30–34]. In the present study, 2D LKMC domains were created from segmented SBF-SEM images by defining a square lattice of uniformly spaced grid points that serve as allowable locations for molecular species. Using a user-defined thresholding criterion (see RESULTS), each grid point was binarized and assigned as either 'pore' or 'cell'; pore locations are accessible to diffusing molecular species, while cell locations are not. The LKMC lattice spacing was chosen to correspond to the pixel resolution of the SBF-SEM images, i.e., 8 nm in both spatial dimensions.

The primary input into an LKMC simulation is a database of all allowable events and their associated rates. In the present work, the only allowable event is a diffusive hop of a molecular entity between neighboring lattice sites. Because the primary purpose of the current study is an assessment of the workflow in Fig 1, the rate of this hopping event is arbitrarily set to correspond to a fixed diffusivity, $D_p = 1 \times 10^{-6}$ cm²/s, a value typical for protein diffusion in water. Molecules are assumed to be tracer particles, i.e., they do not interact with each other and only interact with cell boundaries during diffusion. A molecule is assumed to occupy a single lattice site which corresponds to an upper bound on the effective molecular radius of 8 nm for the default lattice spacing. This is only an upper bound because even if the molecular was assumed to be smaller it could only diffuse to within 8 nm of a cell wall due to the lattice resolution—the issue of lattice resolution will be discussed further in the RESULTS section.

The Next Reaction method [35] was employed to carry out LKMC simulation of diffusion. Here, each event for each particle is assigned a randomly weighted time at which it will occur, given by

$$t_{event} = t_{sys} - \frac{h^2 \log(r)}{D_p} \, ,$$

(1)

where $t_{sys}$ is the time at which $t_{event}$ is assigned ($t_{sys} = 0$ at the start of the simulation), $h$ is the lattice spacing parameter (fixed at 8 nm unless otherwise stated), $r$ is a uniformly distributed random number between 0 and 1. Events are executed in order of increasing $t_{event}$; after each event the system time is updated according to $t_{sys} = t_{event}$, and the $t_{event}$ values for the events associated with the particle that was moved are recalculated. Events that move a particle to a node classified as being of type 'cell' are made inaccessible by setting their $t_{event}$ to an arbitrarily large value. Particles are assumed to be tracers with respect to each other and do not interact.

To address the finite sizes of the domains captured by SBF-SEM images mirror boundary conditions were employed that allow diffusion to proceed beyond the original finite domain. Mirror boundary conditions are a form of periodic boundary conditions but employ reflection to ensure that no discontinuities are encountered at the boundaries to the irregular arrangement of platelets. Although particles rarely diffuse far into the first layer of mirrored domains for the execution times used here, the simulation is constructed such that any particle can theoretically traverse an infinite tiled plane of reflected images. Particle diffusivity was computed via the mean squared displacement, i.e.,

$$MSD(t) = \frac{1}{n} \sum_{i=1}^{n} [\mathbf{r_i}(t) - \mathbf{r_i}(0)]^2$$

(2)

where $n$ is the number of particles in the system, and $\mathbf{r}$ is the position vector of a given particle at a particular time. The diffusion coefficient is directly related to the slope of MSD as a function of time, according to the relationship

$$D = \frac{MSD(t)}{4t}.$$

(3)

## Results

### SBF-SEM image segmentation: Visual accuracy

A total of 110 SBF-SEM images of thrombi, each representing an area of $10 \times 10$ μm$^2$ with a pixel resolution of $8 \times 8$ nm$^2$, were manually labelled by 4 investigators working independently. Manual labelling consisted of identifying platelet boundaries (or other cell boundaries, if present). In certain cases, some subjective judgement was required to locate cell boundaries due to the narrow intercell pore spacing and/or poor image quality. The SBF-SEM images were each classified into one of three bins: (1) sparsely packed configurations ('sparse'), (2) densely packed configurations ('dense'), and (3) densely packed configurations that include a red blood cell ('dense+RBC'). The latter was chosen to ensure that large-scale heterogeneities could be segmented if present in the SBF-SEM images.

To assess the transfer learning capability of Cellpose, we first generated segmentations of images with no additional training. The raw SBF-SEM images used for this purpose were selected from a random pool of 20% of the 110 images from each of the three classification bins. Example segmentations are shown in Fig 3 for both sparse and dense SBF-SEM images. While the base Cellpose model performs well with the sparse domains (Fig 3A), it fails rather obviously in dense regions (Fig 3B,C). In the latter case, many of the platelet boundaries are not identified, resulting in a mask map that contains less than 50% of the platelets by area. Next, the model was fine-tuned using the remaining 80% of the SBF-SEM data. A learning rate of 0.002 was applied for 5000 epochs during the fine-tuning process. To mitigate overfitting, a weight decay of $1 \times 10^{-5}$ was applied in the last 100 epochs of training. As shown in Fig 3, the fine-tuning results in a dramatic improvement of the segmentation accuracy for dense images. After fine-tuning, almost every labelled platelet is identified in the mask map for each of the SBF-SEM images.

The improvement due to the transfer learning approach was assessed quantitively using the Intersection over Union (IoU) metric, which is computed on a pixel-by-pixel basis by comparing a segmented image to its ground truth. Specifically, for each pixel the intersection (nominator) is assigned a value of one if both images identify the pixel as type 'cell', and zero otherwise. The union (denominator) is assigned a value of one if the pixel in either image is of type 'cell'. Consequently, the IoU ranges between zero and one, with a higher IoU indicating greater segmentation accuracy. This metric is particularly suited for evaluating the precision of segmentation where accurate boundary delineation is critical.

The IoU results for both the original and fine-tuned versions of Cellpose are shown in Fig 4 for sparse, dense, and dense+RBC images. For sparse regions, the pre-trained model achieves an IoU of 0.77, which improves to 0.89 after fine-tuning. The improvement is far more significant for dense regions—here the IoU increases from 0.34 to 0.92 for dense domains and from 0.24 to 0.93 for dense+RBC domains. This difference in improvement across different types of domains is not unexpected given the difference in pore geometry. On average, dense regions are characterized by a relatively narrow pore size distribution with a mean pore width of about 40 nm (5 pixels), while sparse regions exhibit pore widths ranging from 40 nm (5 pixels) to 100s of nm (10s of pixels). Quantitative representation of the pore width distributions for different regions are shown in Fig D of S2 Text. Notably, the long tail of the sparse domain distribution shows that many of the pore widths in sparse domains are quite large and therefore easily resolved with the 8 nm resolution of the SEM imaging.

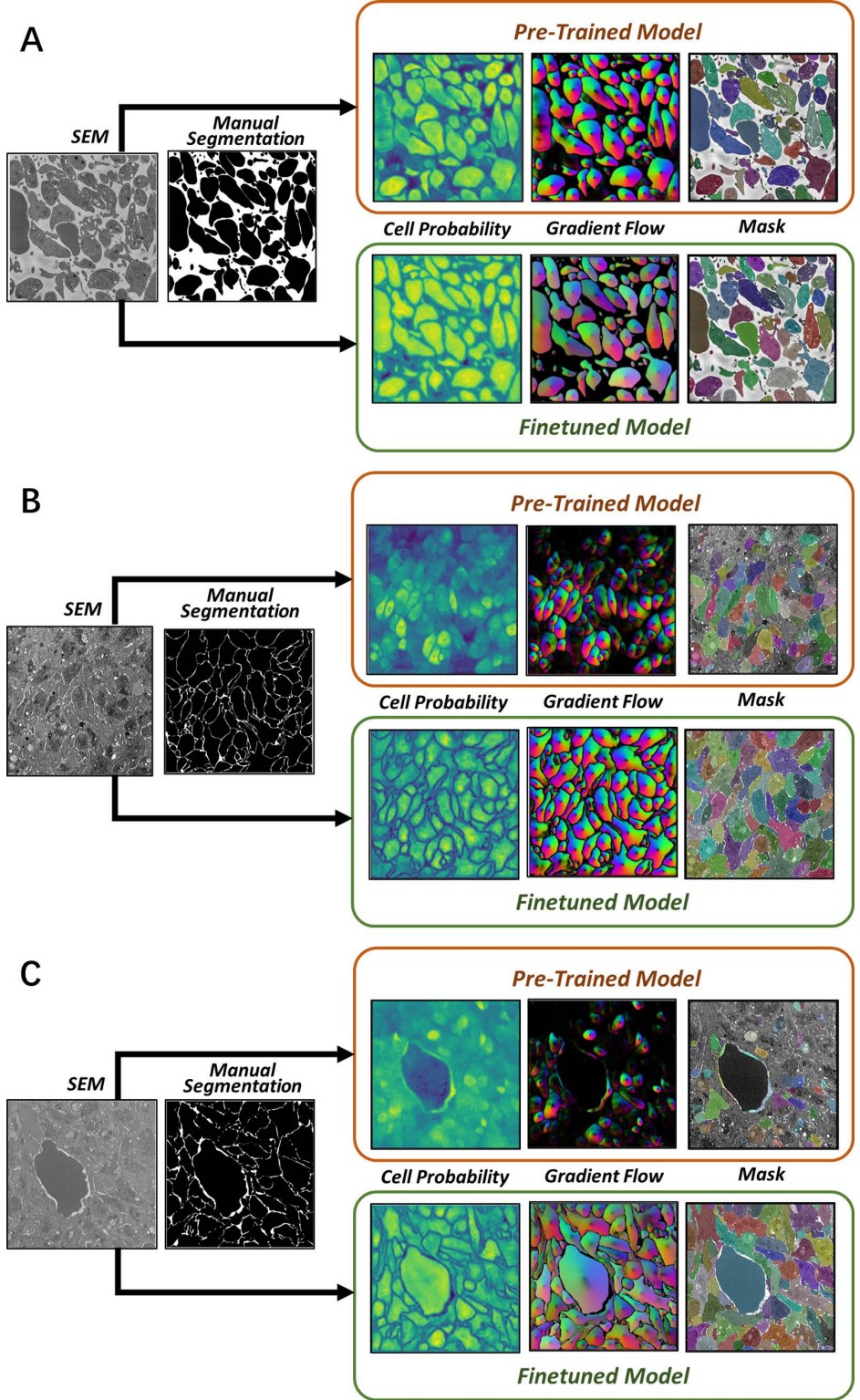

**Fig 3. Example segmentation results.** Segmentation output for **A** sparse images, **B** dense images, and **C** dense+RBC images. For each case, the top row shows the results using the pre-trained Cellpose model, while the bottom row shows results with fine-tuning.

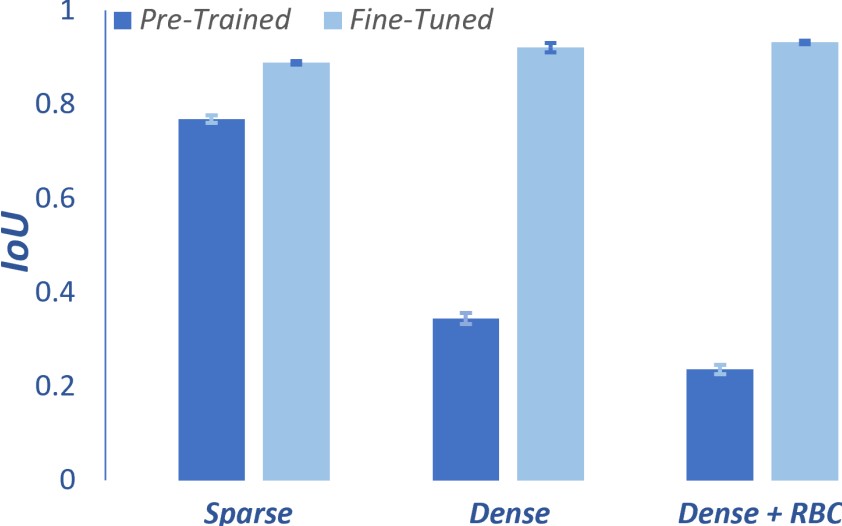

**Fig 4. Segmentation performance assessment.** Mean IoU before and after fine-tuning of Cellpose for sparse, dense, and dense+RBC regions. Error bars represent standard error, $SE = \sigma/\sqrt{n}$, where $\sigma$ is the standard deviation.

Finally, we applied the fine-tuned Cellpose model to a hybrid image that contains both sparse and dense regions. Importantly, this type of image was not used in the fine-tuning of Cellpose. As shown in Fig 5, the fine-tuned model identifies the vast majority of cell boundaries, highlighting its excellent robustness of our segmentation approach.

### Segmented Image-to-LKMC simulation of diffusion: Functional accuracy

While the IoU and related measures are useful for quantifying the extent of overlap between segmented images and the ground truth, they are not necessarily predictive of how well the segmented images serve as computational domains for simulating molecular diffusion and reaction; we refer to this latter notion as *functional accuracy*. There are two key

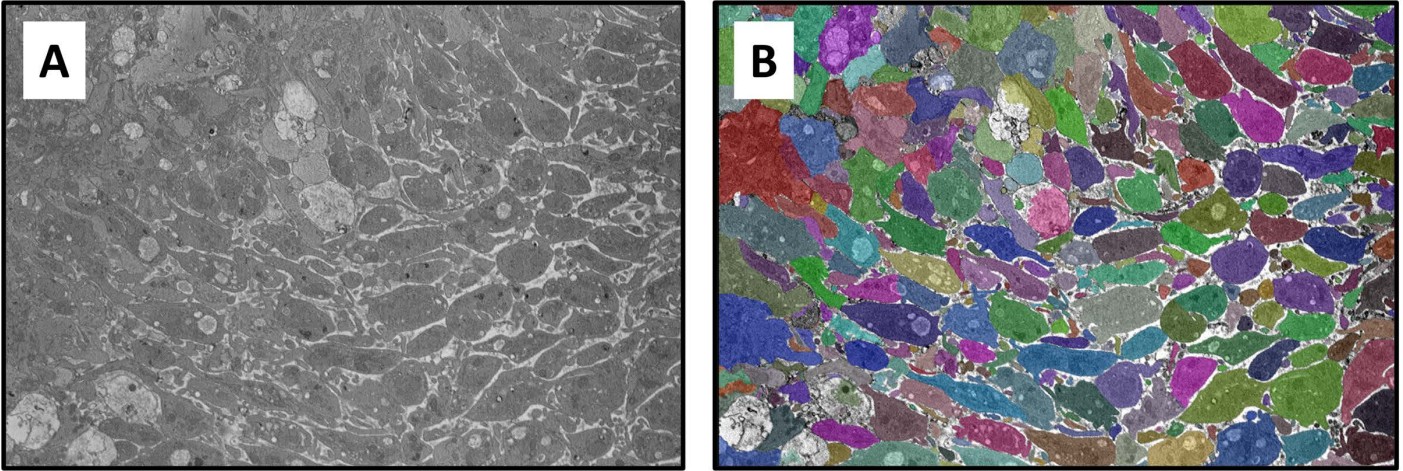

**Fig 5. Segmentation of a hybrid domain containing both sparse and dense regions. A** – raw SEM image, **B** – segmented image with different colored masks representing individual platelets. Images represent a domain size of 23.6 × 16.1 μm².

issues that must be considered in the context of functional accuracy. The first is that molecular diffusion is limited to the inter-platelet pore regions, which represent a rather small percentage of the overall domain, especially in densely packed regions of a thrombus. Consequently, it is possible to obtain good visual accuracy via segmentation, e.g., as measured by IoU, while demonstrating poor functional accuracy due to incomplete resolution of tortuous networks of narrow pore spaces. Secondly, the output of image segmentation must be further processed before it can be used as a computational domain for diffusion simulations. Specifically, cell probability and/or flow field images must be binarized to assign each pixel as corresponding either to a cell or pore; this process requires the specification of a binarization threshold parameter, which is investigated below.

To create an LKMC domain from Cellpose output (either cell probability or gradient flow) we employed the protocol shown in Fig 6. First, the image is converted to grayscale using the NTSC formula [36] with each pixel assigned a scalar brightness value between 0 and 255. In the second step, the grayscale image is binarized according to a *brightness threshold value* (BTV) so that every pixel with a brightness lower than the BTV is assigned to type 'cell', otherwise it is of type 'pore'. Consequently, the greater the BTV, the more porous the domain as shown in Fig 6. Note that manually labelled images, which are used as a reference for evaluating functional accuracy, are effectively already binarized by the labelling process.

A total of 15 images, 5 from each classification bin, were used for investigating the sensitivity of predicted diffusion to the brightness threshold value applied to both cell probability and gradient flow maps; see Fig E-G in S3 Text for raw SEM images, manually segmented domains, and corresponding Cellpose output. A default lattice spacing of 8 nm, corresponding to the native resolution of the SEM imaging, was used for all LKMC simulations and particles were assumed to be tracers with respect to each other. For each of the 15 domains, the diffusivity was computed for a prescribed BTV

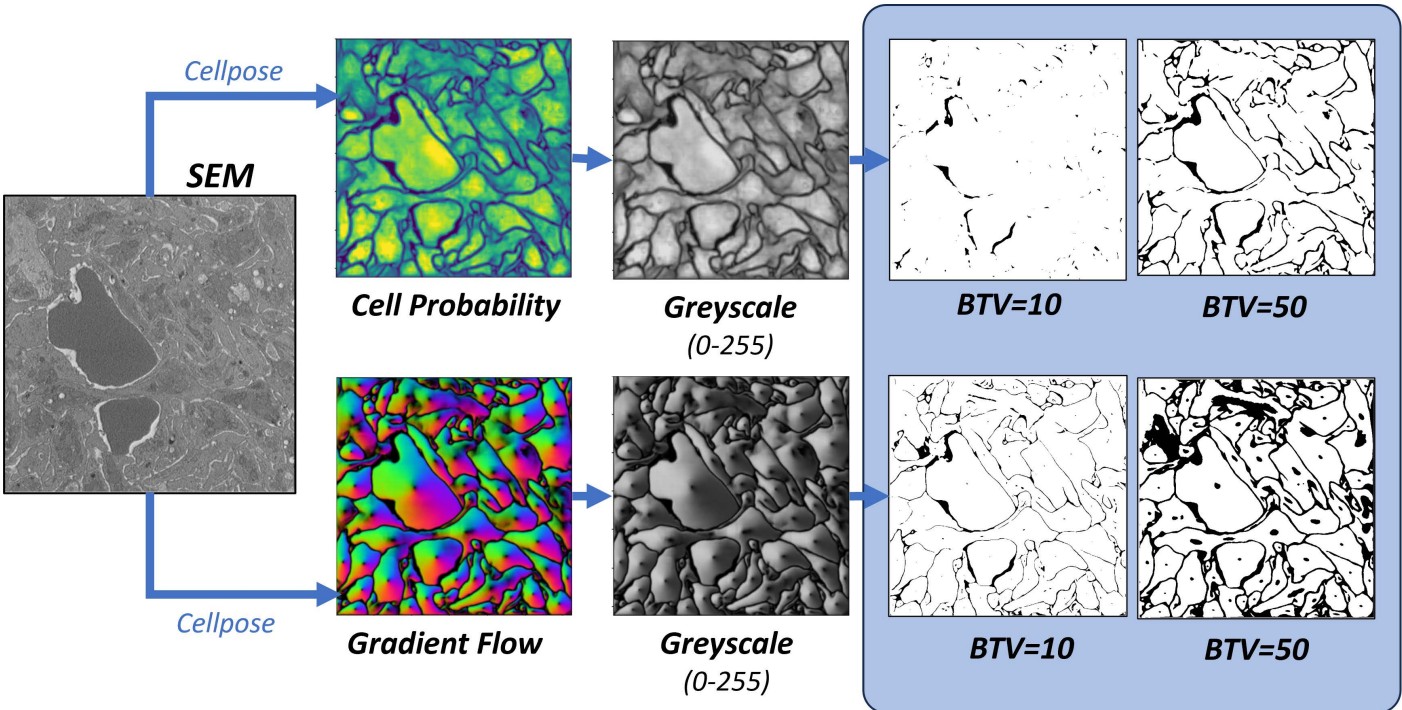

**Fig 6. Binarization of segmented images.** Color cell probability (top) and gradient flow (bottom) maps are converted to grayscale and then binarized using a user-specified brightness threshold parameter (BTV).

value applied to either the cell probability or gradient flow maps. The ground truth diffusivity, $D_{GT}$, was also computed using the manually labelled domain. In Fig 7A, the ratio of the diffusivity obtained in the binarized cell probability relative to the ground truth diffusivity is shown as function of BTV. Each curve represents data averaged over the 5 images of that classification type; the error bars represent the variability of the diffusion coefficient over the 5 domains. For all 3 types of domains, a single BTV for binarizing the cell probability is found that gives a diffusivity equal to the ground truth value. For the dense domains, this value is about 35–40, while for the sparse domains, the optimal BTV is ~110. In other words, the optimal BTV is a function of the image classification—dense images require a higher BTV than sparse ones to reproduce the ground truth diffusivity.

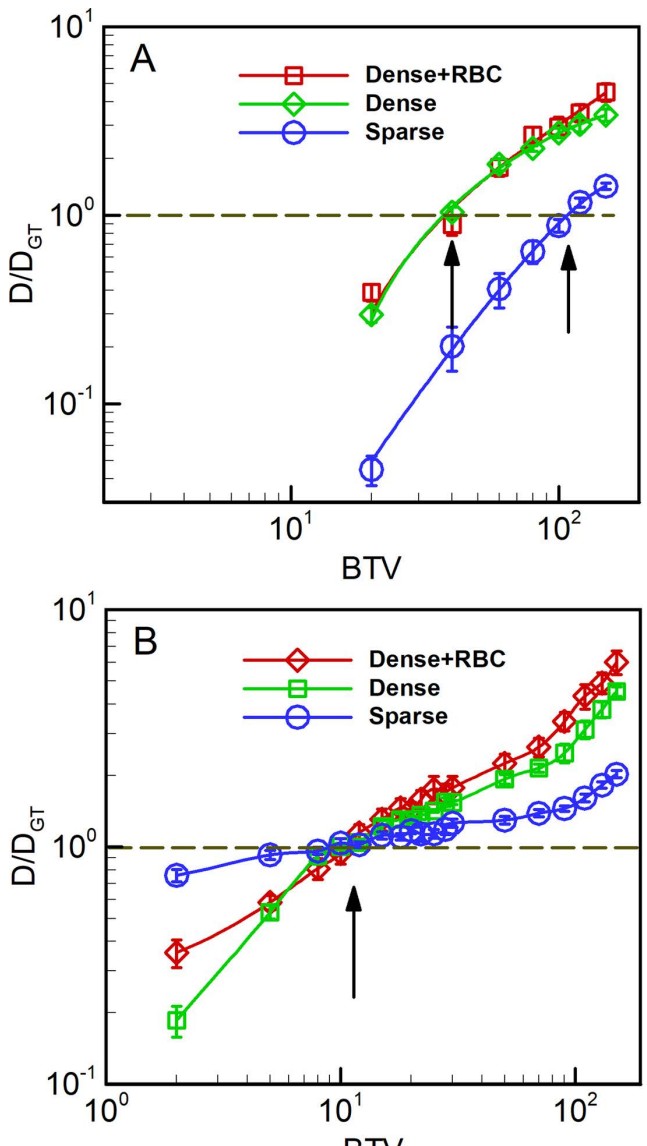

**Fig 7. Predicted diffusivity of 8 nm-radius spherical tracer particles as a function of binarization threshold value (BTV). A** based on the cell probability output, and **B** based on the gradient flow map output, both from Cellpose. Diffusivity values are reported relative to the diffusivity on the corresponding manually labelled domains, $D_{GT}$. Arrows denote locations where $D \approx D_{GT}$ for each case.

PLOS Computational Biology

An equivalent analysis for the corresponding gradient flow maps is shown in Fig 7B. Although the results are qualitatively similar to the cell probability cases, here all three optimal BTVs are essentially equal at ~10. The apparent consistency among optimal BTVs suggests that the gradient flow maps provide a substantially more robust basis from which to generate domains suitable for performing LKMC simulations. While these results do not guarantee that the optimal BTV is completely independent of the packing structure for gradient flow maps, the robustness across the 3 classification types considered here suggests that this is a good approximation. We also considered the impact of LKMC grid resolution (and therefore particle size) on the optimal BTV. As shown in Fig. C in S2 Text, the optimal BTV is only weakly dependent on particle size and the values obtained for the 8 nm lattice spacing may be considered as constants for coagulation-related molecular entities.

### Spatially resolved intrathrombus molecular diffusivity

The segmented and binarized images generated in the previous sections were used in a systematic analysis of molecular diffusion in the hemostatic mass environment. In the following, all LKMC domains were based on the gradient flow output from Cellpose binarized with a BTV of 12. Each $10 \times 10$ µm$^2$ domain was discretized into 81 $2 \times 2$ µm$^2$ subdomains, staggered by 1 µm in the $x$ and $y$ dimensions. LKMC diffusion simulations using the default 8 nm square lattice were executed in each subdomain to obtain a local diffusivity. The diffusivities in regions where the subdomains overlapped were averaged, producing a total of 100 spatially localized diffusivities for each $10 \times 10$ µm$^2$ image; see Fig A in S1 Text for details.

A total of 1000 tracer particles were used for each diffusivity calculation in a given subdomain. Each LKMC simulation was evolved until $t_{sys} = 0.003$ s, which was sufficient to observe the impact of hindrance. For the assumed diffusivity used throughout this study ($D_p = 1 \times 10^{-6}$ cm$^2$/s), this simulation time corresponds to a diffusion length scale of ~1.1 $\mu$m, which is over 50% of the subdomain width. Only the MSD at time $t_{sys} \geq 0.001$s was used to estimate the diffusivity—the short-time MSD ($t_{sys} < 0.001$s) includes contributions from free diffusion in which many particles have not had time to fully explore the hindered environment represented by the pore space. Example MSD plots are shown in S1 Text, Fig. B that highlight the transition from short-time diffusion to long-time diffusion for two different subdomains.

The long-time diffusivities were then used to construct diffusion heatmaps; example heatmaps are shown in Fig 8 for sparse, dense, dense+RBC, and hybrid regions. All diffusivities reported in Fig 8 are normalized by the unhindered diffusivity. As expected, diffusivity values in the sparse domain are generally higher than those in the dense domains. The sparse domain exhibits diffusivities that are about 30–90% of the unhindered values while those in the dense domains are generally in the range of 1–20% of the unhindered value. Less obviously, there is significant variability in the local diffusivity even within a single domain; this type of microscopic variability is usually ignored in coarse-grained models, e.g., those based on continuum descriptions and may have unexpected consequences on the propagation of coagulation reactions within a hemostatic mass.

Next, the porosity of each subdomain across all classes of regions was estimated by computing the fraction of pixels that corresponded to pore space. Generally, the average porosity for sparse regions was ~0.33 while dense regions exhibited porosities around ~0.11. In Fig 9, local diffusivity is plotted against local porosity aggregated over all subdomains generated by Cellpose (blue circles) as well as 30 additional ground truth (i.e., manually labelled) subdomains (orange squares). Interestingly, the data collapses quite convincingly across the entire range of porosity. The scatter, which is substantial, reflects either that (1) porosity is not the only variable that controls local diffusivity, and/or (2) the small subdomains lead to large variations in diffusivity and/or porosity estimation.

## Discussion

The intrathrombus microenvironment is highly complex, exhibiting inhomogeneities at multiple length scales. For example, variations in platelet packing density can lead to large variations in the width and tortuosity of intercellular pore networks containing the plasma medium that transports a multitude of coagulation factors and other molecular moieties.

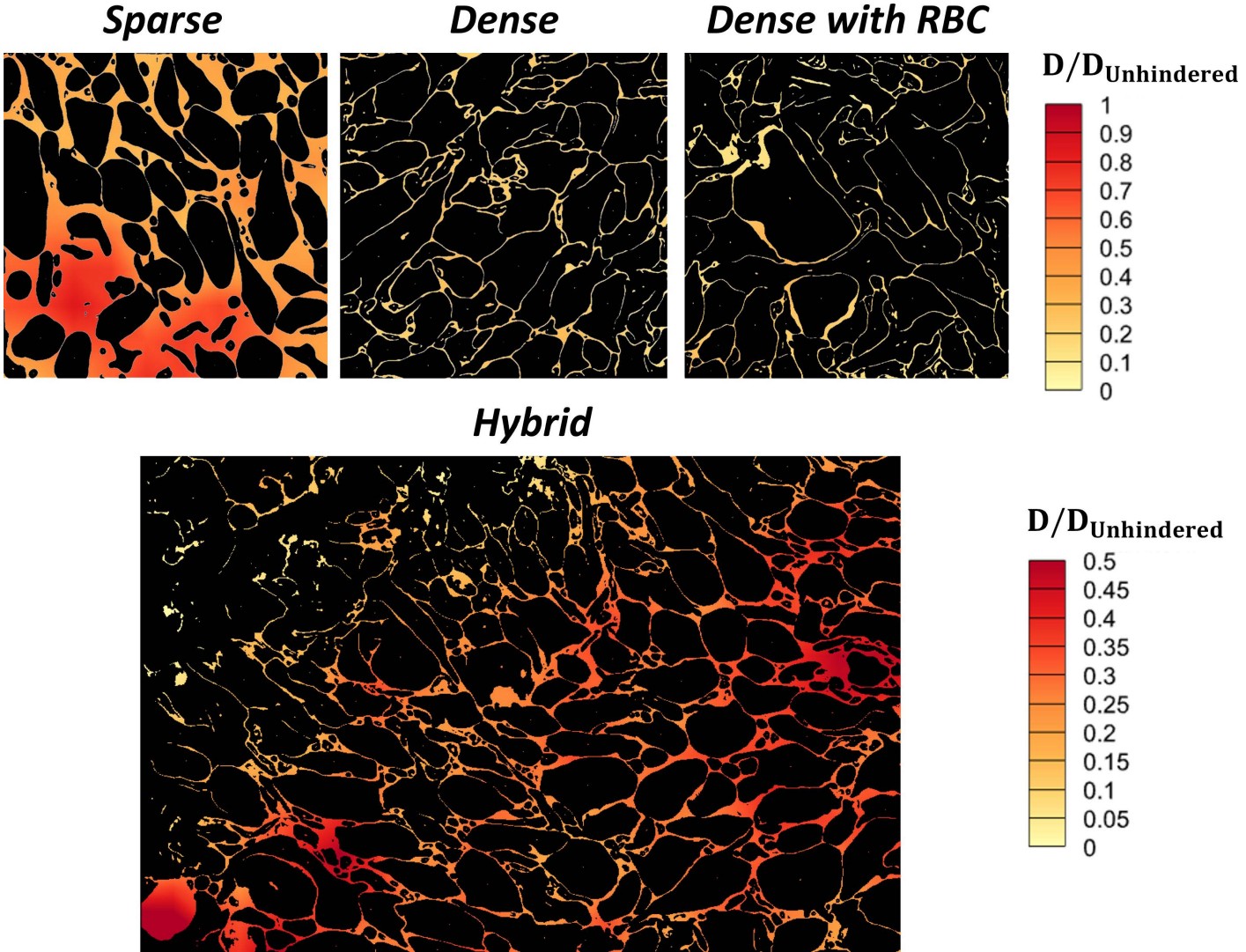

**Fig 8. Normalized diffusion heatmaps for an 8 nm-radius spherical particle.** Top row shows diffusion heatmaps in sparse, dense, and dense+RBC domains, while bottom panel shows diffusivity distribution in a hybrid region of a hemostatic mass. The hybrid domain image represents a larger region and is 23.6 × 16.1 μm², while all others are 10 × 10 μm². All diffusivities are normalized with the unhindered diffusivity in free space.

Plasma-borne coagulation factors comprise a network of biochemical reactions ultimately resulting in thrombin generation. The reactions of this network, most of which occur on cell surfaces, are regulated by molecular transport and can transition between reaction- and transport-limited regimes depending on local conditions [37]. Given heterogeneity in thrombus microstructure, such biophysical effects likely influence not only how much thrombin is generated, but how that thrombin is distributed in time and space as a thrombus evolves, with important consequences for thrombin effector functions. Therefore, understanding the interplay between the microstructure of a hemostatic mass and molecular movement inside of it is a necessary first step towards learning how thrombus microstructure impacts the transmission of coagulation factors and other plasma-borne species.

In this study, we constructed a multistep pipeline for quantitative assessment of diffusional transport in realistic hemostatic mass microarchitectures by combining electron microscopy, artificial intelligence techniques for image

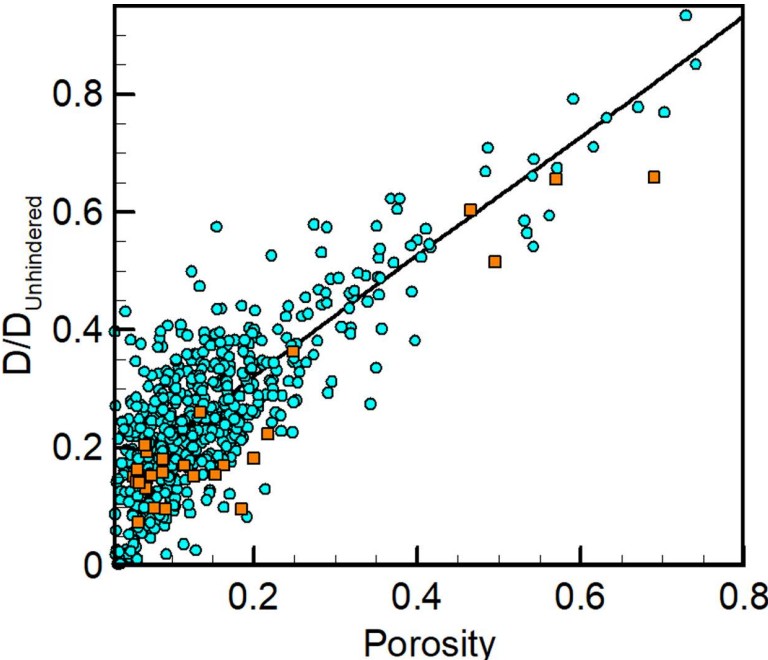

**Fig 9. Diffusivity as a function of hemostatic mass porosity.** Local diffusivity as a function of porosity sampled from $2 \times 2$µm² subdomains taken from both sparse and dense regions. Blue circles represent Cellpose generated domains, orange squares represent ground truth, manually segmented domains.

segmentation, and multiscale particle simulations based on the lattice kinetic Monte Carlo technique. The pipeline required the adaptation, optimization, and integration of existing tools to achieve our goals. Starting from high-resolution images taken from hemostatic thrombi generated in vivo, we first fine-tuned an existing deep learning-based image segmentation package, Cellpose, to automate the segmentation of cells and pore spaces. Using a standard intersection-over-union measure, we found that our fine-tuned Cellpose-based model captured cell boundaries with high fidelity across several different hemostatic mass microenvironments, including samples that exhibit irregularities such as a red blood cell. Crucially, we also introduced the notion of functional accuracy in addition to the standard measures of segmentation quality, which are based only on direct image comparisons. Specifically, we employed quantitative simulations to assess how well the Cellpose-segmented images capture molecular diffusion relative to simulations based on manually segmented ground-truth domains. The resulting framework provides a state-of-the-art approach for translating images of biological microarchitectures into molecular level computational simulations that may be applied in future studies to examine the influence of molecular transport on specific coagulation factors and reactions, as well as other solutes in different biologic tissues.

The optimized workflow was employed to compute spatially resolved diffusion landscapes within various microenvironments. We find that diffusion of spherical particles with sizes similar to those of several coagulation factors is strongly inhibited by the pore architecture to an extent that depends sensitively on the local environment. In domains of sparsely packed platelets the diffusional hinderance is modest, resulting in diffusivities that are typically about 30–90% of the free diffusivity. By contrast, in densely packed platelet regions that are typical of the 'core' regions of hemostatic masses, local diffusivities are as low as 1–20% of the free diffusion value. In transitional regions between these two types of domains, the local diffusivities are observed to vary dramatically over submicron length scales, highlighting the heterogeneity in the transport capacity, and by extension, the ability to transmit chemical signals, across a hemostatic mass.

Notably, relative diffusivities in sparse regions are somewhat higher than those previously inferred for the 'shell' region from homogeneous porous media model fits to experimental measurements [4]. On the other hand, densely packed regions exhibit lower diffusivities than those obtained for the core region [4]. The present study therefore demonstrates a substantially stronger dependence of diffusivity on porosity than previously suggested. However, it is quite likely that porosity alone is insufficient to describe or predict the extent of diffusional hinderance and therefore a more complete description of the pore network would be required to make a quantitative comparison across studies. This hypothesis is supported by the large amount of scatter present in Fig 9, which indicates that a given porosity can exhibit a wide range of diffusivities particularly at low porosities, where the impact of tortuosity and pore network connectivity becomes increasingly important. These results highlight the importance of capturing complex microstructural details when performing quantitative analyses of intrathrombus molecular diffusion.

There are several limitations of the present study that provide opportunities for further investigation. The first is the accuracy of the manual image segmentation that is required for model training. For sparsely packed platelet regions, this is likely to have a small effect because of the clear delineation of cell boundaries. However, densely packed regions are significantly more challenging to process particularly in instances where individual platelets appear to be in contact, requiring some subjective input. Practically, whether a pore is assumed to be completely sealed off by full contact between adjacent platelets or whether a very narrow space exists may result in a large local error but will generally not impact diffusion significantly at larger scales. Additional limitations include the restriction of the present analysis to 2D domains. In practice, SBF-SEM imaging is used to construct 3D domains, but this requires additional image processing in the form of interpolation across a z-stack of 2D SBF-SEM images. Necessarily, for the generation of an LKMC computational domain, interpolation in the third dimension will be subject to larger errors due to the relatively large intervals between adjacent images. Specifically, while the in-plane (x-y) pixel resolution is 8 nm, the spacing between adjacent images is 50 nm for the current dataset (though as low as 25 nm is possible for SBF-SEM imaging). Finally, the present study is limited to diffusion and does not explicitly consider the chemical reactions of the coagulation cascade. While the present results provide a qualitative measure of how diffusional limitations will impact the propagation of coagulation reactions in the hemostatic mass, a complete analysis will require the simultaneous consideration of reaction dynamics. Fortunately, the inclusion of particle-particle and particle-surface reactions is straightforward in LKMC simulations, and this extension will be the focus of future work. We will also consider the impact of heterogeneity in moiety size and shape, which we expect to be significant especially in densely packed regions where the pore widths are comparable to the molecular length scale.

In summary, the present modeling framework represents a foundation for building molecularly resolved simulations of intrathrombus transport and reaction. Key ongoing and future developments include the extension to fully 3D simulations, the inclusion of a reaction network to capture various elements of the coagulation cascade with and without the presence of therapeutic agents, and ultimately a multiscaling scheme to connect multiple localized simulations to address thrombus-wide transport.

## Supporting information

**S1 Text. Fig A in S1 Text.** Binning approach for computing localized diffusivities to generate diffusion heatmaps. **Fig B in S1 Text**. Sample mean-square displacement (MSD) plots for computing localized diffusion coefficients in sparse and dense sub-domains. Green slopes demonstrate the faster rate of diffusion at short time before the effects of the domain are fully felt by particles, while red slopes demonstrate the long-time rate of diffusion. We calculated molecular diffusivity from the MSD curve at t > 0.001 s (black dashed line).
(PDF)

**S2 Text. Fig C in S2 Text.** BTV as a function of particle size/LKMC lattice spacing for different domain types. Optimal BTV values (symbols) represent values at which $D/D_{GT} = 1$. Error bars represent the BTV range for $0.9 \leq D/D_{GT} \leq 1.1$.

Green shaded region at each resolution represents the range of BTV that exhibits $0.9 \leq D/D_{GT} \leq 1.1$ across all domain types, while blue shaded region is the BTV range for which all domain types across all lattice spacings exhibit $0.9 \leq D/D_{GT} \leq 1.1$. **Fig D in S2 Text**. The pore width distribution (in nm) for each domain type (sparse, dense, and dense with RBC).
(PDF)

**S3 Text. Fig E in S3 Text.** The five sparse domains used in our analysis of the optimal BTV of Cellpose's segmentation outputs. Four versions of each domain are pictured: the original SEM image, the manually segmented ground truth, Cellpose's gradient flow output, and Cellpose's cell probability output. **Fig F in S3 Text**. The five dense domains used in our analysis of the optimal BTV of Cellpose's segmentation outputs. Four versions of each domain are pictured: the original SEM image, the manually segmented ground truth, Cellpose's gradient flow output, and Cellpose's cell probability output. **Fig G in S3 Text**. The five dense+RBC domains used in our analysis of the optimal BTV of Cellpose's segmentation outputs. Four versions of each domain are pictured: the original SEM image, the manually segmented ground truth, Cellpose's gradient flow output, and Cellpose's cell probability output.
(PDF)

## Author contributions

**Conceptualization:** Maurizio Tomaiuolo, Timothy J. Stalker, Talid Sinno.

**Data curation:** Maurizio Tomaiuolo, Timothy J. Stalker.

**Formal analysis:** Catherine House, Ziyi Huang, Kaushik N. Shankar.

**Funding acquisition:** Maurizio Tomaiuolo, Timothy J. Stalker, Talid Sinno.

**Investigation:** Catherine House, Ziyi Huang, Kaushik N. Shankar, Sandra J. Young, Meghan E. Roberts.

**Methodology:** Catherine House, Ziyi Huang, Kaushik N. Shankar, Scott L. Diamond, Maurizio Tomaiuolo, Timothy J. Stalker, Lu Lu, Talid Sinno.

**Project administration:** Maurizio Tomaiuolo, Talid Sinno.

**Resources:** Scott L. Diamond, Maurizio Tomaiuolo, Timothy J. Stalker, Talid Sinno.

**Software:** Catherine House, Ziyi Huang, Kaushik N. Shankar.

**Supervision:** Maurizio Tomaiuolo, Timothy J. Stalker, Lu Lu, Talid Sinno.

**Validation:** Catherine House, Ziyi Huang.

**Visualization:** Catherine House, Ziyi Huang.

**Writing – original draft:** Catherine House, Maurizio Tomaiuolo, Talid Sinno.

**Writing – review & editing:** Catherine House, Ziyi Huang, Scott L. Diamond, Maurizio Tomaiuolo, Timothy J. Stalker, Lu Lu, Talid Sinno.

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
