## [Decision Letter · Decision Letter 0]

PCOMPBIOL-D-25-00226

From imaging to computational domains for physics-driven molecular biology simulations: Hindered diffusion in platelet masses

PLOS Computational Biology

Dear Dr. Sinno,

Thank you for submitting your manuscript to PLOS Computational Biology. After careful consideration, we feel that it has merit but does not fully meet PLOS Computational Biology's publication criteria as it currently stands. Therefore, we invite you to submit a revised version of the manuscript that addresses the points raised during the review process.

Please submit your revised manuscript within 30 days Jun 30 2025 11:59PM. If you will need more time than this to complete your revisions, please reply to this message or contact the journal office at ploscompbiol@plos.org. Please include the following items when submitting your revised manuscript:

We look forward to receiving your revised manuscript.

Kind regards,

Shugang Zhang

Academic Editor

PLOS Computational Biology

Marc Birtwistle

Section Editor

PLOS Computational Biology

**Journal Requirements:**

1) Please upload all main figures as separate Figure files in .tif or .eps format. For more information about how to convert and format your figure files please see our guidelines: 

2) We have noticed that you have uploaded Supporting Information files, but you have not included a list of legends. Please add a full list of legends for your Supporting Information files after the references list.

3) Some material included in your submission may be copyrighted. According to PLOSu2019s copyright policy, authors who use figures or other material (e.g., graphics, clipart, maps) from another author or copyright holder must demonstrate or obtain permission to publish this material under the Creative Commons Attribution 4.0 International (CC BY 4.0) License used by PLOS journals. Please closely review the details of PLOSu2019s copyright requirements here: PLOS Licenses and Copyright. If you need to request permissions from a copyright holder, you may use PLOS's Copyright Content Permission form.

Potential Copyright Issues:

i) Figure 1. Please confirm whether you drew the images / clip-art within the figure panels by hand. If you did not draw the images, please provide (a) a link to the source of the images or icons and their license / terms of use; or (b) written permission from the copyright holder to publish the images or icons under our CC BY 4.0 license. Alternatively, you may replace the images with open source alternatives. See these open source resources you may use to replace images / clip-art:

4) We note that your Data Availability Statement is currently as follows: "All relevant data are within the manuscript and its Supporting Information files.". Please confirm at this time whether or not your submission contains all raw data required to replicate the results of your study. Authors must share the “minimal data set” for their submission. PLOS defines the minimal data set to consist of the data required to replicate all study findings reported in the article, as well as related metadata and methods (https://journals.plos.org/plosone/s/data-availability#loc-minimal-data-set-definition).

6) Please ensure that the funders and grant numbers match between the Financial Disclosure field and the Funding Information tab in your submission form. Note that the funders and grant numbers must be provided in the same order in both places as well.

**Reviewers' comments:**

Reviewer's Responses to Questions

Reviewer #1: This is a well written paper that describes a computational pipeline to quantify diffusion of particles in 2D images that are segmented into cellular matter and surrounding pores. The value of this approach is its technique to quantify estimates of diffusivity in in-vivo imaged data. The authors acknowledge all limitations that came to my mind as I was reading the manuscript. I think this manuscript is worthy of publication after a few minor revisions.

1. The abstract misleads that "large stacks" of 2D images are being used to computationally model a 3D thrombus. in fact, this is why I agreed to review this manuscript. Please better communicate your actual approach.

2. The motivation and background section is superb.

3. Please clearly indicate in the text (earlier than the limitations section) that your difusion analyses are only for 2D planes.

4. 1000 particles diffusing for 0.003 sec seems really coarse and fast. Are the 1000 particle cross the entire domain or for each subdomain? Please clarify.

5. Please give an estimate (on average) of how many pixels wide are the typical pore widths in each of the three image types.

6. Please provide typical CPU times and resources required to perform the CNN training and well as run the heat maps.

7. In Fig. 9, can you add and/or highlight which data points are associated with the ground truth segmented images? The reason is: the reader may question how necessary is it to include the additional CNN segmented images if the 5-15 manually segmented images essentially provide the same linear fit?

8. The Discussion section is weak. The first 3 paragraphs read like an extended abstract, rehashing things already stated in more detail in the Results section. The latter paragraphs have good discussion content. Please decide how to resolve this. A short Discussion section is fine.

9. Please beef up the "so what?" aspect in your discussion. How does this work generalize to something of future relevance for thrombosis modeling? Perhaps discuss future work and/or forecasting how this work can be synergistically coupled to other modeling approaches.

Reviewer #2: I believe that, this is the original work of the authors. The quality of the manuscript is good and fulfill the requirements of the journal. This is an interesting observation and there is very little in the literature on this topi. The contents are presented in a clear and narrative manner, which is easy to understand for readers with various background.

Weaknesses:

- Not much has been said about the importance of this study, and this is the weakness of the article.

- There is a lack of a general discussion to summarize the results. There is no proper discussion based on the results obtained in the article.

Reviewer #3: The manuscript describes a pipelined process to reconstruct from SBF-SEM images the initial conditions (computational domain) for numerical simulations. A lattice kinetic MC method is used to simulate molecule diffusion on these reconstructed comput. domains. Although the result of hindered diffusion of molecules through cell packed domains is previously known, using deep learning to help image analysis is quite interesting.

**Have the authors made all data and (if applicable) computational code underlying the findings in their manuscript fully available?**

Reviewer #1: Yes

Reviewer #2: None

Reviewer #3: **No: ** The raw image, the lattice kinetic MC code are not available.

PLOS authors have the option to publish the peer review history of their article (what does this mean? ). If published, this will include your full peer review and any attached files.

**Do you want your identity to be public for this peer review?** For information about this choice, including consent withdrawal, please see our Privacy Policy .

Reviewer #1: No

Reviewer #2: **Yes: ** Reza Razaghi

Reviewer #3: No

**Figure resubmission:**
---

## [Decision Letter · Decision Letter 1]

Dear Dr. Sinno,

We are pleased to inform you that your manuscript 'From imaging to computational domains for physics-driven molecular biology simulations: Hindered diffusion in platelet masses' has been provisionally accepted for publication in PLOS Computational Biology.

Best regards,

Shugang Zhang

Academic Editor

PLOS Computational Biology

Marc Birtwistle

Section Editor

PLOS Computational Biology

Reviewer's Responses to Questions

**Comments to the Authors:**

Reviewer #2: The authors have taken reviewers comments seriously and the manuscript is much improved as a result.

**Have the authors made all data and (if applicable) computational code underlying the findings in their manuscript fully available?**

Reviewer #2: Yes

PLOS authors have the option to publish the peer review history of their article (what does this mean? ). If published, this will include your full peer review and any attached files.

**Do you want your identity to be public for this peer review?** For information about this choice, including consent withdrawal, please see our Privacy Policy .

Reviewer #2: No

---

## [Editor Report · Acceptance letter]

PCOMPBIOL-D-25-00226R1

From imaging to computational domains for physics-driven molecular biology simulations: Hindered diffusion in platelet masses

Dear Dr Sinno,

I am pleased to inform you that your manuscript has been formally accepted for publication in PLOS Computational Biology. Your manuscript is now with our production department and you will be notified of the publication date in due course.

With kind regards,

Zsofia Freund
